# Characterizing and Exploiting Tumor Microenvironments to Optimize Treatment Outcomes

**DOI:** 10.3390/cancers13225752

**Published:** 2021-11-17

**Authors:** Emma H. Allott, Kellie Dean, Tracy Robson, Claire Meaney

**Affiliations:** 1Patrick G. Johnston Centre for Cancer Research, Queen’s University Belfast, Belfast BT9 7AE, UK; 2Department of Histopathology and Morbid Anatomy, Trinity Translational Medicine Institute, Trinity College Dublin, D08 HD53 Dublin, Ireland; 3School of Biochemistry and Cell Biology, University College Cork, T12 XF62 Cork, Ireland; k.dean@ucc.ie; 4School of Pharmacy and Biomolecular Sciences, Royal College of Surgeons Ireland, D02 YN77 Dublin, Ireland; tracyrobson@rcsi.ie; 5Department of Mechanical and Automobile Engineering, Limerick Institute of Technology, V94 EC5T Limerick, Ireland; Claire.Meaney@lit.ie

**Keywords:** microenvironment, immune system, microbiome, preclinical models, organoids, genetically engineered mouse models

## Abstract

**Simple Summary:**

The Irish Association for Cancer Research (IACR) held its 57th annual conference from the 24–26 March 2021 in a virtual format due to the ongoing pandemic. This report provides a summary overview of the work presented at the conference, which had a particular focus on the tumor microenvironment. Tumors do not develop and grow in isolation, but rather within the context of their surrounding environment. The work presented at the conference outlined the complexity of the tumor microenvironment and highlighted several ways in which it influences tumor growth and progression. Moreover, the tumor microenvironment was discussed as a potential target for new cancer treatments. Traditionally, laboratory research has focused on the tumor only, but conference speakers highlighted the importance of modeling the surrounding microenvironment to gain a more physiologically relevant view of tumorigenesis. Finally, conference attendees heard from the patient’s perspective regarding the development of novel targeted therapies.

**Abstract:**

Our understanding of cancer initiation, progression, and treatment is continually progressing through dedicated research achieved through laboratory investigation, clinical trials, and patient engagement. The importance and complexity of the microenvironment and its role in tumor development and behavior is pivotal to the understanding of tumor growth and the best course of treatment. The 57th Irish Association for Cancer Research (IACR) Annual Conference collected key researchers, clinicians, and patient advocates together to highlight and discuss the recognized importance of the microenvironment and treatment advances in cancer. In this article, we describe the key components of the microenvironment that influence tumor development and treatment, including the microbiome, metabolism, and immune response and the progress of preclinical models to reflect these complex environments. From a psycho-social oncology perspective, we highlight expert opinion and data on the process of shared decision-making in the context of emerging cancer treatments.

## 1. Introduction

Cancers develop in complex tissue microenvironments that they depend upon for sustained growth, invasion, and metastasis. The tumor microenvironment has immense complexity and diversity, comprising a variety of cell types, including innate and adaptive immune cells, fibroblasts, as well as blood and lymphatic vasculature. This collection of cells contained within the extracellular matrix is commonly referred to as the stroma. Within a normal context, the microenvironment provides critical signaling to maintain the tissue architecture and suppress malignant growth. However, in carcinogenesis, the tumor can co-opt and communicate with the microenvironment to promote and support its growth [1].

Once the importance of the tumor microenvironment in cancer growth and metastasis was appreciated, it was proposed as an attractive therapeutic target given its lower genetic complexity relative to tumor cells [2]. However, we are now appreciating the tremendous complexity of the tumor microenvironment and only beginning to characterize how it varies across different organ types according to tumor genetic make-up. Therefore, much work needs to be done before novel therapies can be identified and translated to the clinic. Careful consideration and selection of preclinical models are needed to understand how best to target the tumor microenvironment, as traditional models have omitted the microenvironment.

In this paper, we highlight research presented at the 57th Irish Association for Cancer Research, which informs our understanding of novel aspects of the tumor microenvironment (Table A1). We also consider expert opinions on how the tumor microenvironment could be targeted therapeutically, and what preclinical models are needed to better understand this complex relationship to ensure that new therapies developed in the lab ultimately reach the clinic. Finally, we conclude by discussing the patient perspective on the development and implementation of novel targeted cancer therapies.

## 2. Stromal Cell-Mediated Immunosuppression in Colorectal Cancer (CRC)

Dr. Oliver Treacy from the National University of Ireland Galway presented data on the role of sialylation (a form of glycosylation) in stromal cell-mediated immunosuppression in colorectal cancer. Dr. Treacy’s presentation focussed on the immunomodulatory role of sialic acid sugar-carrying glycans, sialoglycans, expressed on stromal cells in the CRC tumor microenvironment. Sialoglycans are recognized by sialic acid-binding immunoglobulin-like lectins (Siglecs), a family of immunomodulatory receptors, which are analogous to the immune checkpoint inhibitor, programmed cell death protein 1 (PD-1) [3].

Initial studies using human bone marrow-derived mesenchymal stromal cells (MSCs), pre-cursors to cancer-associated fibroblasts (CAFs) [4], demonstrated that MSCs expressed higher levels of sialic acid than the CRC cell lines HCT116 and HT29. The next objective was to confirm/validate this finding and, using CRC patient-derived CAFs and tumor-adjacent normal-associated fibroblasts (NAFs), Treacy and colleagues showed that CAFs expressed higher levels of sialic acid with a particular affinity for a specific type of linkage, α2,6. They expanded upon this finding by screening CAFs and NAFs for expression of specific Siglec ligands, namely, Siglec 7/9 ligands using Siglec 7/9 Fc chimeras. Their results showed that while Siglec 7 ligand expression was comparable between CAFs and NAFs, Siglec 9 ligand expression was clearly increased on CAFs. As they were interested in investigating the effects of CAF sialylation on CAF-mediated suppression of T cells, they next confirmed that CD4 and CD8 T cells isolated from the peripheral blood of healthy donors expressed the inhibitory Siglec receptors Siglec 7 and Siglec 9, which are more commonly associated with monocytes and natural killer (NK) cells [5]. Co-culture assays revealed that CAFs induced significantly higher frequencies of Siglec 7 and Siglec 9-expressing CD8 T cells, as well as PD-1-expressing CD8 T cells, compared to NAFs. Inhibition of sialyltransferases, the enzymes responsible for adding newly synthesized sialic acid to underlying glycans, reversed these CAF-induced effects. Interestingly, sialyltransferase inhibition had no observed effects on T cells co-cultured with NAFs.

Overall, this work shows that targeting stromal cell sialylation can reverse immune cell suppression and reactivate exhausted T cells. These novel data support a rationale for the assessment of stromal cell sialylation and Siglec ligand expression to better stratify patients for immunotherapeutic combination treatments that aim to reactivate exhausted T cells in stromal-enriched tumor microenvironments.

## 3. Microbiota and Cancer: What? So What? Now What?

While not traditionally considered part of the microenvironment, microbiota have now been observed within internal organs previously thought to be sterile, and these populations have the potential to be explored as cancer biomarkers and/or targets for novel treatments. Professor Mark Tangney from the University College Cork, Ireland opened the ‘Microbiota and Response to Therapy’ session describing some of his lab’s work on intratumoral bacteria. His talk described a decade of preclinical and clinical studies on the characterization and exploitation of bacterial growth within tumors. Prof. Tangney described witnessing a change in the focus of the literature on the exploration of bacterial presence in tumors, from an earlier mindset of specific bacteria as pre-existing causative agents of cancer to explorations on the possibility of their presence being subsequently opportunistic inhabitants [6]. The Tangney lab has a long history in the field of systemic administration of engineered bacteria in tumor therapy [7], and his presentation described the facets of the tumor environment, which uniquely supports selective bacterial growth-immune suppression, leaky vasculature, and regions of low oxygen and rich bacterial nutrient availability [8]. The capability in his lab, coupled with the rise of microbiome sequencing technology, led to their study of patient tumor samples for potential bacterial presence, and his group pioneered describing the presence of a broad microbiome within both malignant and non-malignant breast tissue in 2014 [9].

Since then, multiple labs have reported similar findings in various tumor types, but the need for improvements in methodology became apparent due to the difficult nature of ‘low bacterial biomass’ sample types involved and the need to eliminate potential sources of contamination. Prof. Tangney presented details from several recent papers from his group aimed at tackling this problem, through improvements in bioinformatics [10], sequencing [11], and sample processing aspects [12,13], in addition to improved protocols for sample acquisition [14].

The rationale for the title of his talk quickly became apparent, as he described ‘What’ as the characterization of the tumor microbiome [15], and ‘So What’ as studies on potential implications of bacterial presence, such as his group’s finding in 2015 that such bacteria enzymatically altered commonly used chemotherapeutic drugs in mouse models [16]. Finally, an example of ‘Now What’ was given with a description of exploitation of such findings in a biomarker context, describing a new study from his group where the development of a machine-learning strategy enabled the team to use the bacterial signature of fresh tissue biopsies to predict their malignancy status [14].

## 4. Exploring and Therapeutically Exploiting the Tumor Microenvironment

Professor Johanna Joyce, the University of Lausanne and the Ludwig Institute of Cancer Research, Lausanne, Switzerland presented research from her lab on the characterization and therapeutic targeting of the tumor microenvironment (TME) in brain cancers. While some brain tumors originate in the brain as primary tumors, others are metastases from other primary sites, most commonly breast, lung, and skin (melanoma) [17]. Given this notable heterogeneity of brain cancers, the Joyce lab seeks to understand whether the brain TME is shaped in a tissue-specific or a disease-specific way, as well as how to overcome the immunosuppressive nature of this microenvironment. The lab takes a mouse-to-human approach, integrating data from human samples and genetically engineered mouse models (GEMMs), characterizing the TME in patients and therapeutically targeting it in mice. Clinical samples include malignant brain tumors (both primary and metastatic), non-malignant brain tumors (rapid autopsy or benign conditions such as epilepsy), in addition to matched blood.

The Joyce lab recently published their findings in Cell in 2020 [18], showing a diverse immune cell landscape across different types of brain cancers shaped in a disease-specific way. In the brain, tumor-associated macrophages (TAMs) are either resident microglia (MG) or infiltrating monocyte-derived macrophages (MDMs). Joyce and colleagues reported a significant shift in the ratio of MG to MDMs between high-grade and low-grade gliomas, and a low abundance of T cells, in keeping with the immunologically ‘cold’ nature of glioma and the corresponding modest effects of immunotherapies in this tumor type [19]. Moreover, within brain metastases, the composition of the immune compartment of the TME differed by primary site, with CD4+ and CD8+ T cells showing the highest abundance in melanoma brain metastases, whereas breast metastases showed the highest neutrophil infiltration. Integrating gene expression data with immune cell phenotyping showed T cell exhaustion and activation of immune suppression signaling pathways in brain metastases.

Joyce and colleagues propose that efforts to reprogram or re-educate TAMs rather than simply depleting them will likely be the most successful [20]. To that end, the lab has used colony-stimulating factor 1 receptor (CSF-1R) inhibitor to experimentally target TAMs. This approach induces tumor dormancy in GEMM models of high-grade glioblastoma, but approximately half of the animals go on to develop a tumor recurrence. RNA sequencing data from these recurrences support a role for compensatory signaling pathways, including insulin-like growth factor 1 (IGF-1) and phosphoinositide 3-kinase (PI3K), which, if targeted alongside CSF-1R treatment, reduce the development of resistant recurrences [21]. Moreover, treatment with CSF-1R shows evidence of increased neuronal markers in tumor cells, pointing to neuronal mimicry as a potential resistance mechanism. This highlights the potential compensatory pathways that should be targeted to improve therapeutic response. Future work will focus on longitudinal monitoring of the TME response to treatment, alongside combination therapies to address the evolving compensatory mechanisms as they arise.

## 5. Crossing the Valley of Death: The Need for Better Preclinical Mouse Tumor Models

Research presented by Professor Hellmut Augustin, Professor of Vascular Biology at the Medical Faculty Mannheim of Heidelberg University, Germany, highlighted the need for better preclinical mouse models, particularly for studying metastasis. Currently, only 1 in 20 drugs entering clinical trials in oncology are approved, and this is one of the lowest rates of approval of any disease [22]. Better preclinical mouse models are critical to facilitate higher rates of approval in oncology.

A systematic review and meta-analysis of in vivo mouse models from Professor Augustin’s group reported that 80% of all preclinical mouse research is performed using cell-line-derived models, which are generally less representative of human cancers [23]. Moreover, he discussed how the interpretation of tumor growth curves in mice could be misleading, with most studies reporting results at the experimental endpoint (i.e., tumor harvest) and without considering growth curves at different stages. Professor Augustin demonstrated that the log transformation of growth curves could more clearly highlight the point at which the curves diverge between treatment groups. An early divergence in growth curves could indicate that therapy may be better suited to earlier-stage disease. If considering endpoint data alone, this early divergence in growth becomes amplified over time and could result in a misleadingly large difference in tumor size at harvest. Finally, with respect to the study of metastasis, Professor Augustin’s group found that only a tiny proportion of in vivo studies of metastasis involved resection of the primary tumor. As such, mouse models thought not to be metastatic may simply not be metastatic within the timeframe that it takes the primary tumor to kill the mouse.

Professor Augustin discussed how to compromise on selecting appropriate mouse models, finding a balance between complexity and reproducibility. He proposed two strategies employed by his lab. The first is a genetically modified syngraft model, incorporating surgical resection of the primary tumor to study metastasis. This model involves selecting a nodule or ‘fragment’ of a multi-focal tumor from a genetically engineered mouse model (GEMM) and grafting it onto a syngeneic mouse. Primary tumors are surgically resected and biobanked (including multi-omic characterization), allowing the mouse to develop multi-focal metastases. The second approach is the creation of a focal GEMM using electroporation, inducing tumor formation locally (preferably orthotopically), followed by surgical resection to facilitate the study of subsequent metastasis. These approaches reflect the biology of GEMM but are more versatile and more easily reproducible. Importantly for the overarching theme of this year’s conference, the use of these model systems also preserves the contribution of the TME to tumor growth and metastasis.

In summary, Professor Augustin highlighted the need to move beyond the use of endpoints to study tumor progression instead of leveraging growth curves to identify the optimal timing of therapeutic interventions. He recommends a gradual phasing out of tumor cell line experiments in mice and greater efforts to mimic the human course of disease in the study of metastasis, including primary tumor resection. Through these approaches to achieve better modeling of human disease in vivo, we will overcome a key bottleneck in cancer research and improve approval rates for new cancer treatments.

## 6. Stem Cell-Based Organoids in Human Disease

An alternative approach to modeling tumors ex vivo was presented by Professor Hans Clevers, Professor of Molecular Genetics at Utrecht University and Principal Investigator at the Hubrecht Institute (KNAW) and the Princess Máxima Center for Pediatric Oncology and Oncode Investigator. He began by acknowledging that our current understanding of cancer initiation and progression has been very much influenced by the technologies available in a laboratory setting. While the use of 2D monocultures of cancer cell lines has enabled many discoveries, it is clear that cancer cell lines cannot replicate the complexity of a patient’s tumor [24]. Professor Clevers discussed his work in developing organoids derived from normal and diseased tissues, like the intestine.

Professor Clevers is a world-renowned expert in stem cell biology. Human organs have a niche of stem cells that allow for continued repair, and the stem cells in the crypts of intestinal villi continually replace the epithelial cells at the surface. These cells can be isolated from the intestinal crypts and expanded under certain defined laboratory conditions to create 3D organoids or “mini-guts”, which contain all six cell lineages present in the gut [25]. Tumor organoids can contain non-neoplastic tissue components in addition to the neoplastic cells, thereby enabling the study of the tumor cells within the native microenvironment. As highlighted above, the role of microbiota in both healthy organ function as well as diseased states, including cancer, is increasingly recognized. Professor Clevers’ lab has demonstrated that organoid models can be used to study the role of intestinal microbiota by injecting microbes into the organoid lumen [26]. Under the direction of Professor Clevers, normal and tumor tissue samples from patients have been used to create “living biobanks” of well-characterized organoid lines that recapitulate histological and genetic features of individual tumors from the breast, colon, kidney, and stomach [27,28,29,30]. This allows researchers to test directly how various drug therapies will affect an individual’s tumor without the use of animal models.

Professor Clevers’ use of organoids as cancer models will fundamentally change the field. Having multiple cell layers interacting in a 3D environment more closely resembles the tumor as it exists within a patient. Better models will lead to a better understanding of cancer cell behavior and ultimately more tailored treatments to exquisitely target an individual’s specific tumor. Cancer organoids derived from an individual’s cancer will allow the promise of personalized medicine to become a reality. 

## 7. The Psycho-Social Impact of Cancer

With the development of novel therapies, the patient experience of targeted approaches, generally with fewer side effects than traditional chemotherapy, requires consideration. Dr. Tina Hickey, a psychologist, presented personal reflections on the implications of diverging from the common narrative of breast cancer treatment while undergoing mastectomy and endocrine therapy. She reported that the narrative of breast cancer that is most often seen in the media still portrays chemotherapy as an intrinsic part of breast cancer treatment. Yet, in recent years, more than half of all female breast cancer patients in Ireland did not undergo chemotherapy as part of their treatment [National Cancer Registry Ireland (NCRI), 2019], and this proportion is likely to increase given medical advances. While her personal experience involved mastectomy and endocrine therapy, these experiences may also apply to patients receiving existing and novel targeted therapies. She reported data from a small qualitative study discussing the impact of ‘Imposter Syndrome’ on patients in the non-chemotherapy treatment group, which she argued can disrupt help-seeking, based on a belief that ‘others have it much harder than me’.

Dr. Hickey reports that there is currently a risk that patients whose distress falls below extreme levels, or whose need for support is urgent to make timely decisions about surgical and treatment options, are quite likely to have to seek out and assemble their own psycho-social supports, which introduces systematic inequalities based on resources, education, geography, and other factors. Reflecting on her own experience, Dr. Hickey observed that timely access to psycho-oncology counseling in the short period before her surgery was critical in her case; without a small number of sessions then, she believed her treatment decisions regarding surgery and reconstruction would have been less emotionally informed, probably resulting in less satisfaction with her surgical outcomes, and therefore involving subsequent surgery and greater overall distress, risk, and cost.

Speaking from her own experience, she noted that information on available supports for individuals diagnosed with cancer is highly fragmented across charities and services, and she called for the provision of a comprehensive Central Information Hub with information and links to all supports. She also advocated the establishment of ‘Wellness Centres’ where cancer survivors could access information and treatment for a range of post-acute treatment conditions such as lymphedema, fatigue, pain, and problems regarding relationships, sexuality, return to work, and fears of recurrence. Dr. Hickey identified cost-effective supports and interventions that helped significantly during her recovery, reporting the benefits she gained from learning pain management techniques post-surgery through a mindfulness app rather than relying on painkillers and visualization techniques to help in coping. She identified the support she found in a peer-group of individuals living with cancer and taking part in interventions, such as those around counseling, physiotherapy, exercise, and self-help materials, as crucial in her recovery. She stressed the importance of allowing participants sufficient time to share their experience and not structuring the intervention delivery and materials so rigidly that they seem to ‘crowd out’ the vital interpersonal support that patients find in such groups. Furthermore, she argued that it is crucial that such interventions show sensitivity and respect to their participants in allowing space for patients to raise concerns that matter to them, rather than the program topics only.

Overall, Dr. Hickey stressed that psycho-oncology needs to become more inclusive of patient groups with diverse illness narratives, whose distress lies on a continuum of severity, and whose needs may be served at different times by resource-intensive or more cost-effective, diverse provisions. She argued that a model of care that normalizes timely access to a wider range of psycho-social supports would pay dividends in detecting and preventing more severe distress later.

## 8. Future Research Directions

The conference concluded with the IACR Award for Outstanding Contribution to Cancer Medicine and Research being awarded to Professor Mark Lawler, Associate Pro-Vice-Chancellor, Professor of Digital Health, and Chair in Translational Cancer Genomics at Queen’s University Belfast (QUB), Northern Ireland. Professor Lawler is Associate Director of Health Data Research Wales-Northern Ireland, which is driving innovative precision medicine and public health approaches using big data. In his acceptance speech, he outlined his path from cancer researcher in precision medicine to a proponent of big data usage to address cancer inequities and tackle the negative implications of COVID-19 on cancer diagnoses and treatment pathways. Furthermore, Professor Lawler and colleagues have championed the coming together of researchers from both sides of the border to bring together expertise from Northern Ireland and the Republic of Ireland to address the common enemy that is cancer. As such, there is currently a collaborative push across the island of Ireland for researchers both North and South of the border to come together to understand the role of the microenvironment in cancer and how it could be therapeutically targeted to improve cancer outcomes in Ireland and worldwide.

## 9. Conclusions

The research presented at the 57th IACR Annual Conference highlighted the important role of the tumor microenvironment in cancer progression and treatment response. Identification and development of appropriate model systems will be key for studying the mechanisms of tumor progression in the context of the microenvironment. This, in turn, will enable the development of novel therapies targeting the tumor microenvironment to improve patient outcomes.

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
