# Peer review of "Characterizing and Exploiting Tumor Microenvironments to Optimize Treatment Outcomes"

_cancers, 2021, doi:10.3390/cancers13225752_

Round 1

Reviewer 1 Report

This well written manuscript reports a summary overview of the work presented at the 57th annual conference of the Irish association for cancer research, dedicated to tumor microenvironment, a major issue in cancer research nowadays. 

Among the six presentations, the authors chose very interesting subjects on novel aspects of tumor microenvironment, and adapted preclinical models such as tumor organoids.

The last example is dedicated to the patient perspective for novel targeted cancer therapies, targeting the tumor microenvironment for example. However, precise examples of such therapies should be clearly identified. 

Author Response

Point 1: This well written manuscript reports a summary overview of the work presented at the 57th annual conference of the Irish association for cancer research, dedicated to tumor microenvironment, a major issue in cancer research nowadays. 

Among the six presentations, the authors chose very interesting subjects on novel aspects of tumor microenvironment, and adapted preclinical models such as tumor organoids.

The last example is dedicated to the patient perspective for novel targeted cancer therapies, targeting the tumor microenvironment for example. However, precise examples of such therapies should be clearly identified.

Response 1: We appreciate the opportunity to clarify this point. The speaker in this section was not specifically referring to novel microenvironment-targeted therapies but rather to existing non-chemotherapy treatments; in her case, surgery and endocrine therapy. To clarify this, we have removed the reference to therapies targeting the microenvironment in the opening line of this section, and inserted the following text:  

“While her personal experience involved mastectomy and endocrine therapy, these experiences may also apply to patients receiving both existing and novel targeted therapies”.

Reviewer 2 Report

In this review article, authors summarize many tumor microenvironment topics that discussed in the 57th annual conference of Irish Association for Cancer Research (IACR) held at 2021, March. Author emphasized on stromal cells and microbiota on cancer progress. They further discussed therapeutic targeting for tumor metastasis in the tumor microenvironment, such as TAM. They mentioned the importance of establishing tumor metastasis model and tumor organoids for pre-clinical study to improve the development of cancer treatments. In the final part, they offered another point of view which is psycho-oncology counselling. This is a very good summary report with update citation.

Minor point: Just need to re-consider the section title of 8, is part far from conclusion or summary of the conference, should have the other paragraph for conclusion section. The remaining section 8 needs to consider to changing into future aspect of conference or vise versa.

Author Response

Point 1: In this review article, authors summarize many tumor microenvironment topics that discussed in the 57th annual conference of Irish Association for Cancer Research (IACR) held at 2021, March. Author emphasized on stromal cells and microbiota on cancer progress. They further discussed therapeutic targeting for tumor metastasis in the tumor microenvironment, such as TAM. They mentioned the importance of establishing tumor metastasis model and tumor organoids for pre-clinical study to improve the development of cancer treatments. In the final part, they offered another point of view which is psycho-oncology counselling. This is a very good summary report with update citation.

Minor point: Just need to re-consider the section title of 8, is part far from conclusion or summary of the conference, should have the other paragraph for conclusion section. The remaining section 8 needs to consider to changing into future aspect of conference or vise versa.

Response 1: We appreciate this suggestion. We have replaced the title of section 8 with “Future research directions”, as suggested. We have added a new section 9, “Conclusions”, with the following text:

  1. Conclusions

The research presented at the 57th IACR Annual Conference highlighted the important role of the tumor microenvironment in cancer progression and treatment response. Identification and development of appropriate model systems will be key for studying the mechanisms of tumor progression in the context of the microenvironment. This, in turn, will enable the development of novel therapies targeting the tumor microenvironment to improve patient outcomes.